# Dopamine transients in the ventral striatum provide evidence for average-reward reinforcement learning

## Abstract

Agents in real environments need to organize their behavior over a wide range of time scales. This might be achieved by reinforcement learning (RL) algorithms employing a spectrum of discount factors. Neural evidence for this idea includes recordings of dopamine (DA) release transients, which appear to reflect shorter time horizons in dorsal striatum and much longer horizons in ventral striatum (VS). However, this also presents a challenge, because with very long time horizons all states have similar, large values, impeding learning. Prior theoretical work has therefore proposed algorithms, including average-reward RL, that segregate out the large shared component of value. Here we compare temporal-difference reward prediction errors derived from recurrent neural network models (RNNs) to rat VS DA transients measured in three behavioral tasks. We show that using average-reward RL to train RNNs can provide an improved match to VS DA, compared to using discounting alone. We further find that the activity dynamics in RNNs trained with average-reward RL readily encodes key decision variables such as recent reward history, in a task-specific manner. The functional alignment between DA dynamics and average-reward RL may offer new insights into neural mechanisms of learning and decision-making.

## 1 Introduction

A seminal connection between neuroscience and machine learning has been the interpretation of dopamine (DA) signals as conveying temporal-difference (TD) reward prediction errors (RPEs) of reinforcement learning (RL) (Schultz et al., 1997; Sutton & Barto, 2018). Within this framework, agents adapt their behavior to optimize an estimate of aggregate future rewards—typically discounted over time. Encoding of RPE has been observed both in DA cell firing and release transients, especially in the striatum (Hart et al., 2014; Mohebi et al., 2019). In Mohebi et al. (2024), DA transients in three different striatal regions were recorded throughout an extended period of training, from the initial cue exposure to the successful learning of associations between different cues and rewards. DA transients in dorsolateral and dorsomedial striatum were effectively modeled using RL with discount factors corresponding to time scales from seconds to tens of seconds. However, DA in the ventral striatum (VS) appeared to reflect a much longer time scale of reward estimation (e.g. on the order of 1000 s) according to a variety of measures across multiple behavioral tasks. Notably, it was reported that DA transients in the VS (unlike other striatal regions) required extended training to distinguish between different cues, and also showed positive responses to the cue that is never followed by reward. This was interpreted as reflecting the inherent difficulty in distinguishing values associated with cues when—over a long time horizon that encompasses many trials—all cues are followed by a large number of rewards.

More generally, using long time horizons that encompass many episodes of experience can result in all states having similar, large values (Naik et al., 2024). This can tax representational accuracy and impede learning. To avoid this, various approaches have been proposed that segregate out the component of value that is shared across states—notably average-reward RL (Mahadevan, 1996; Dewanto et al., 2020). In neuroscience, average-reward TD learning has been proposed to model behavior and DA signals during classical conditioning (Daw & Touretzky, 2000; 2002; Daw et al., 2006) and foraging (Shuvaev et al., 2020). In this work, we propose that VS DA transients in

particular may reflect RPEs from an algorithm similar to average-reward RL, that estimates values relative to a shared reference value. Using average-reward RL we train recurrent neural networks (RNNs) to perform both Pavlovian conditioning and operant tasks, and show that network RPE signals resemble DA transients recorded in the VS of rats performing the same tasks. Average-reward RL avoids a difficulty encountered by models that discount with long time horizons, namely values that continue to grow despite extended training.

A second goal of this work was to examine the internal processes by which RNNs are able to effectively estimate values and make adaptive choices in these simulated behavioral tasks. We find that the network dynamics of RNNs trained using average-reward RL appropriately track the decision variables—such as reward rate—that are useful for the specific task. This enables "meta-learning" (Wang et al., 2018) whereby the network can make adaptive adjustments in output based on recent experience without requiring changes in connection weights. These observations support the possibility that algorithms resembling average-reward RL may be employed by the brain, especially by circuits including VS that help guide behavior over extended time scales.

## 2 METHODS AND MATERIALS

### 2.1 BEHAVIORAL TASKS AND THEIR IMPLEMENTATIONS

We consider three behavioral tasks: two Pavlovian conditioning tasks and one operant bandit task, as described in detail in previous work (Mohebi et al., 2019; 2024). For the conditioning task with probabilistic rewards, one of three possible auditory cues indicated a reward delivery after a fixed delay with probability 75%, 25% or 0%, respectively. Each cue lasted for 2.6 s, then after a 0.5 s gap could be followed by a reward delivery click in a rewarded trial indicating the reward was ready to be collected. The inter-trial interval (ITI) period lasted between 15-30 s. There were also unpredicted reward deliveries with the same frequency as the other cues. For the conditioning task with multiple delays, one of three possible cues indicated a 75% chance of reward delivery after a cue-reward delay of 0.6, 3 or 12 s, respectively. In the bandit task, a trial started when the center port light turned on (light-on). After the rat poked into the center port (center-in), it had to keep holding there for a variable period of 0.5-1.5 s till a go-cue occurred and lights at the two adjacent ports turned on. The rat then freely chose one of the two adjacent ports (side-in), each with a predefined probability of triggering a click indicating reward delivery at the food port. The reward probabilities for the two side-ports included all combinations of 0.1, 0.5 and 0.9, and remained constant during blocks of 40-60 trials. After a block ended, the reward probabilities changed randomly without notification. ITIs ranged from 5-10 s.

In modeling the two conditioning tasks, we followed the task implementation in Mohebi et al. (2024). There were two actions "poke" and "non-poke", and a small action cost was introduced after taking the action "poke". The cues included both one-hot and overlapping portions of their representations. In modeling the bandit task, the actions included "non-engaged", "engage", poking into each of the three ports, poking into the food port, and "movement". During the ITI period of each trial, the action started with "non-engaged", then switched to "engage" before poking to the middle port ("center-in") and starting the holding period. Transition from one action to another required passing through at least one "movement" action.

For the conditioning task with probabilistic rewards, the inputs to the RNN represented environmental signals, which were composed of the background input, the cues, and reward delivery click (Mohebi et al., 2024). The background input has 3 dimensions (dim) with content 1. Each cue representation has 20 dim, with 17 dim as overlapping features with content 1 and 3 dim as one-hot representation for three different cues. The reward click was 1 dim with content 5. In total, the inputs were 24 dim. The same input representation was used for the conditioning task with multiple delays, although in the two tasks the three cues have different meanings.

For the bandit task, the inputs to the RNN included three components. The environmental signals included the light-on signals for the three ports (3 dim), the go-cue (1 dim), reward click (1 dim) and background input (3 dim). The remaining two components were composed of the reward received at the last timestep (1 dim) and one-hot representation of the action taken at the last timestep (7 dim, the size of the action space). In total the inputs were 16 dim.

DA signals (as reported in Mohebi et al. (2019; 2024)) were measured using fiber photometry of the fluorescent sensor dLight, in the rat VS. For the conditioning task with probabilistic rewards, recordings started from the beginning of training. For the conditioning task with multiple delays and the bandit task, recordings started after the rats had fully learned the task.

## 2.2 NETWORK MODEL

We built RNNs with an actor-critic structure (Mnih et al., 2016; Wang et al., 2018). The network is composed of LSTM (long short-term memory) units (Hochreiter & Schmidhuber, 1997). The loss function included three terms, policy loss, value loss and entropy term (Mnih et al., 2016):

$$L^\theta = L^P(\theta) + \beta_V L^V(\theta) - \beta_E L^E(\theta) \ . \tag{1}$$

The network was trained with APO (Ma et al., 2021), which is a generalization of proximal policy optimization (PPO) (Schulman et al., 2017) to the average reward case. During training, the average reward was updated as

$$\hat{\eta} \leftarrow (1 - \alpha)\hat{\eta} + \alpha \frac{1}{T} \sum_{t=0}^{T-1} r_t \ , \tag{2}$$

where $r_t$ is the reward received at time step $t$, and $T$ is the sequence length.

The policy loss has the following form

$$L_t^P(\theta) = \min(\rho_t \hat{A}_t, \text{clip}(\rho_t, 1 - \epsilon, 1 + \epsilon)\hat{A}_t) \ , \tag{3}$$

where $\rho_t = \frac{\pi_\theta(a_t|s_t)}{\pi_{\theta_{old}}(a_t|s_t)}$ was the probability ratio clipped with a parameter $\epsilon$. The advantage $\hat{A}_t$ was the generalized advantage estimator (GAE) (Schulman et al., 2015) of the TD error $\delta_t = r_{t+1} - \hat{\eta} + \hat{V}_{t+1} - \hat{V}_t$. The value loss was given by

$$L_t^V(\theta) = \frac{1}{2}(\bar{r}_t - \nu b - V_t)^2 \ , \tag{4}$$

where $\bar{r}_t = r_t - \hat{\eta} + \hat{V}_{t+1}$ is the TD target in average-reward RL, $b = \frac{1}{T} \sum_{t=0}^{T-1} \hat{V}_t$ is the mean value over sequence length $T$ and $\nu$ is a parameter. The $b$ term was added to ensure the mean of values was 0 as expected in the average-reward driven RL (Ma et al., 2021). For the bandit task, we used $\bar{r}_t = \hat{A}_t + \hat{V}_t$ to better account for the GAE influence. $L^e$ is the entropy of the probability distribution for taking each action, added to encourage exploration (Mnih et al., 2016). The network weights were updated using the Adam method (Kingma & Ba, 2014). In the conditioning task, a long continuous sequence of 500 s was used to mimic the session structure in rat experiments and to capture the development of RPEs during training (in above equations we dropped the batch index to represent this setup). For the bandit task, the network was trained using parallel environments (batch size 64) for better sampling efficiency, with a sequence length of 40 s. When analyzing network activity in the bandit task we considered only the trained network with already optimized choosing behavior. Parameters used for the conditioning and bandit tasks are shown in Table 1. For the bandit task, we used the implementation in Huang et al. (2022) and extended it to the average-reward RL (Ma et al., 2021). The simulation was performed on a CPU cluster.

For training RNN with a long time horizon and reward centering, we also used the algorithm provided in Ma et al. (2021). The discount factor $\gamma$ and time horizon $\tau$ are related by $\gamma = e^{-dt/\tau}$, where $dt$ is the time step size. For $dt = 0.1$ s, the time horizon $\tau = 1000$ s as suggested for the VS in Mohebi et al. (2024) corresponds to $\gamma = 0.9999$. It is well known that when $\gamma \to 1$, the accumulated discounted rewards $\eta_{\pi,\gamma}$ and average reward $\eta_\pi$ under a given policy $\pi$ in a Markov decision process (MDP) was given by $\eta_{\pi,\gamma} \to \frac{1}{1-\gamma}\eta_\pi$, i.e., the accumulated reward diverges as $\gamma \to 1$ (Puterman, 1994). Therefore, when subtracting the reward at each time step by $\eta_\pi$, i.e, "reward centering" as proposed in Naik et al. (2024), the value function under a policy $\pi$ could be defined as

$$V_{\pi,\gamma}(s) = \mathbb{E}_{\omega \sim \pi} \left[ \sum_{t=0}^{\infty} \gamma^t(r(s_t, a_t) - \eta_\pi) \mid s_0 = s \right] \ , \tag{5}$$

where $s_t$, $a_t$, and $r(s_t, a_t)$ are the state, action and reward at time step $t$, and the expectation is over the sampled path $\omega = (a_0, s_1, a_1, s_2, a_2, ...)$ from the policy $\pi$. The value function defined in

Table 1: Network model and training parameters

| Name | Conditioning tasks | Bandit task |
|---|---|---|
| time step size $dt$ (s) | 0.1 | 0.1 |
| seq length $T$ (s) | 500 | 40 |
| num LSTM units | 32 | 64 |
| input dim | 24 | 16 |
| action space | 2 | 7 |
| action cost | -0.0006 | 0 |
| learning rate | 0.0005 | 0.0002 |
| $\lambda$ in GAE | 0.98 | 0.95 |
| $\epsilon$ in clip | 0.1 | 0.1 |
| $\beta_V$ | 0.8 | 0.5 |
| $\beta_E$ | 0.001 | 0.05 |
| $\alpha$ | 0.1 | 0.1 |
| $\nu$ | 0.5 | 0.5 |

Eq. (5) was well behaved for any $\gamma$ and has zero mean at state $s$ (Cao, 2007; Ma et al., 2021; Naik et al., 2024). Specifically, Ma et al. (2021) provided a unified trust region theory for both $\gamma < 1$ (Schulman, 2015; Achiam et al., 2017) and $\gamma = 1$ (Zhang & Ross, 2021). We used algorithm from Ma et al. (2021) for both average-reward RL and discount RL with reward centering.

In the two conditioning tasks, the network was first updated 500 times with cue presentations turned off (pretraining period). In the bandit task, the network was first trained to learn center-in, side-in and food-port-in actions (i.e., receiving rewards after learning each of those procedures) before learning the full task. This helped with procedural learning, resembling the sequential steps used to train rats to perform the same task. The source code for our simulations is available at this anonymous GitHub repository.

### 2.3 ADDITIONAL ANALYSIS DETAILS

The RPE to be compared with the DA signal was defined as

$$RPE_t = r_t - \hat{\eta} + \hat{V}_t - \hat{V}_{t-1} \ , \tag{6}$$

where $r_t$ is the reward received at time step $t$ and $\hat{V}_t$ is the estimated value at time step $t$.

In the bandit task, we defined a reward rate $\rho$ using a leaky integrator:

$$\rho_t = (1 - \alpha_0)\rho_{t-1} + \alpha_0 r_t \ , \tag{7}$$

where $r_t$ is the reward received at time step $t$ and $\alpha_0 = 0.001$. We divided all the trials used in evaluation (200 blocks with the first block excluded) into three quantiles (High, Med, and Low) based on the reward rate at light-on for each trial.

## 3 RESULTS

### 3.1 RPEs AND VALUES IN THE CONDITIONING TASKS

As DA dynamics in the VS appeared to reflect estimates of reward over very long time scales, we considered the possibility that the underlying algorithm might actually implement average-reward RL, with an infinite time horizon. We trained an actor-critic RNN with an average-reward driven policy gradient method as developed in Ma et al. (2021). We found that RPEs at cue onset (Fig. 1c, d) showed similar patterns to the DA transients from rats (Fig. 1a, b) and to the RPEs from an RNN model with a long time horizon (Fig. 5c, d in Appendix). Both types of RNN model displayed a notable feature previously reported for VS DA dynamics: the delayed discrimination of different cues across training (Mohebi et al., 2024). This slow discrimination reflected the slow development of distinct trajectories of unit activity for different cues (Fig. 2a). Both types of RNN also effectively

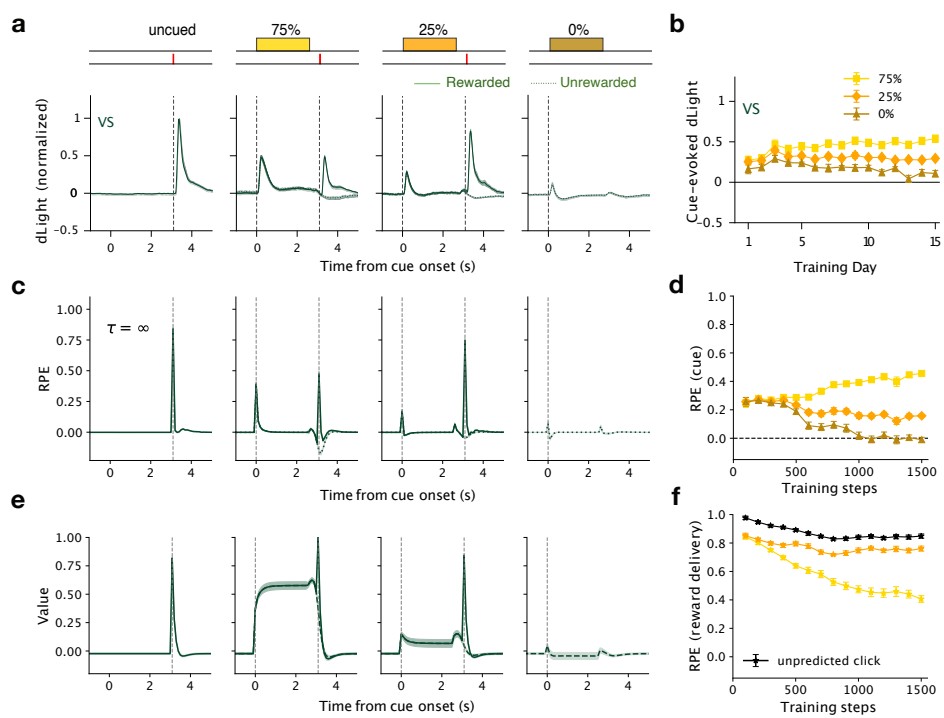

Figure 1: The RNN model trained using average-reward RL reproduced the DA transients at VS in the conditioning task with probabilistic rewards. **a**. Top, cues (colors indicate different pitches for auditory pip trains) and potential reward click times (red lines) for the four intermixed trial types (three with cues, and unpredicted rewards). Bottom, DA transients in the VS, averaged over the last three days of training. **b**. Development of DA transients at cue onset over training. **c**. RPEs for each trial type from the network model at training step 1000. **d**. Development of RPEs at cue onset time for the three cues over training. **e**. Value functions for each trial type at training step 1000. **f**. Development of RPEs at reward delivery click time in rewarded trials over training. Data in **a**-**b** were adopted from Mohebi et al. (2024). Data in **c**-**f** were presented as mean ± s.e.m, averaged over 20 seeds with each run including 1000 trials.

reproduced the RPE patterns at cue onset seen for DA (Mohebi et al., 2024). However, the long-horizon model showed an ever-increasing value function during training (Fig. 2b). As anticipated, the average-reward model resolved this problem by subtracting the average-reward from the value function (Eq. 5 with $\gamma = 1$), which makes the value function zero mean (Fig. 1e, Fig. 2b). At the time of the reward click, VS DA also encoded RPE: greater DA release was observed when reward was received following the 25% cue compared to the 75% cue (Fig. 1a) (Mohebi et al., 2024). This pattern was reproduced by the average-reward model (Fig. 1c, d), but not the long-horizon model (Fig. 5c in Appendix).

While average-reward RL avoids discounting altogether, an alternative proposed solution retains long-horizon discounting but simply subtracts the average reward from the value function ("reward centering", as proposed in (Naik et al., 2024)). With reward centering, an RNN with long horizon produced similar RPE and value function results to the model trained with average-reward RL (Fig. 6 in Appendix). This is as expected, since $\gamma = 0.9999$ (correspondingly, $\tau = 1000$ s with $dt = 0.1$ s) is very close to the limit 1 (infinite horizon), and in this limit average-reward RL and discount RL with reward centering are identical (Ma et al., 2021).

In the conditioning task with multiple delays, the average-reward RL model reproduced the experimentally observed DA scaling patterns (Mohebi et al., 2024): the cue response decreased with the

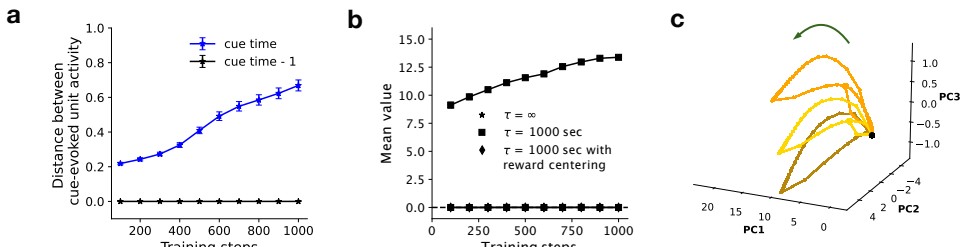

Figure 2: Unit activities and value functions with training and trajectories in the PCA space in the RNN model trained using average-reward RL. **a**. Development of Euclidean distance between unit activities of cue pairs over training, to measure discrimination between cues. Distances were measured right after cue onset and one step before cue onset, and averaged over all pairs of the three cue types. **b**. Development of mean values for average-reward RL ($\tau = \infty$) over training. For comparison, values from discount RL ($\tau = 1000$ s) without and with reward centering were also showed. In **a**-**b**, data were presented as mean $\pm$ s.e.m, averaged over 20 seeds with each run including 1000 trials. **c**. Trajectories of unit activities in PCA space for all the cued trials in an example run (same color scheme as Fig. 1). The black dot represents an attractor right before cue onset. Arrow direction indicates the subsequent time flow. Each trajectory moves away from the attractor during the cue, then changes direction sharply at cue offset; a subsequent split in each trajectory reflects whether the reward click occurred or not.

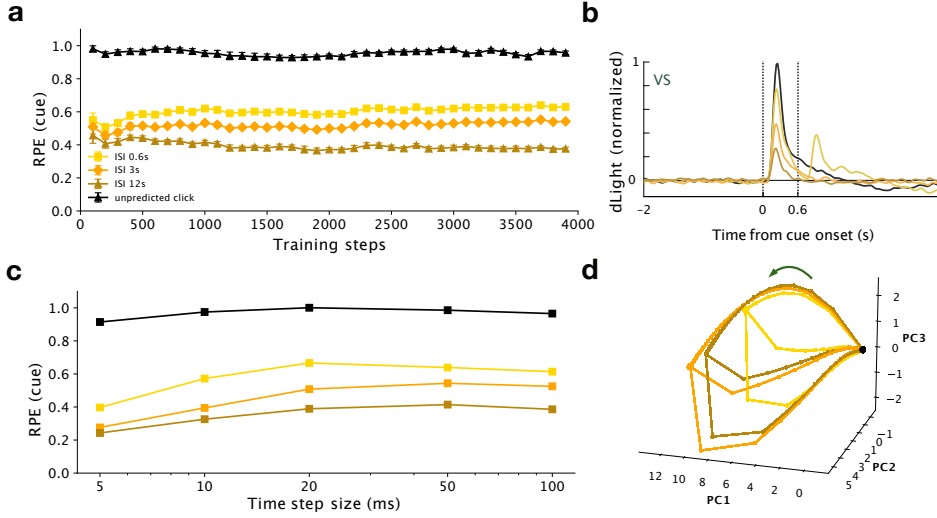

Figure 3: RPEs and population dynamics for the RNN model trained using average-reward RL in the conditioning task variant with multiple delays. **a**. Development of RPEs at cue onset time with training for three trial types with different inter-stimulus intervals (ISIs) and unpredicted reward clicks. **b**. Experimental DA activity from well-trained rats. Data adopted from Mohebi et al. (2024). **c**. RPEs at cue onset showed only weak dependence on time step size. Data were from training step 3000. In **a** and **c**, data were presented as mean $\pm$ s.e.m, averaged over 10 seeds with each run including 1000 trials. **d**. Trajectories of unit activities in PCA space for all the cued trials in an example run. The black dot represents an attractor right before cue onset. Arrow direction indicates the time flow after cue onset.

increase of reward delay (Fig. 3a, b; Fig. 8 in Appendix). We found that this scaling held for a wide range of model time resolutions (Fig. 3c).

### 3.2 POPULATION DYNAMICS IN THE CONDITIONING TASKS

After training in the probabilistic rewards task, RNN population activity followed distinct trajectories (visualized in the space of the first three principal components; Fig. 2c). These highly stereotyped trajectories were determined only by the cue type and whether a reward was delivered. This was appropriate for this task, for which trials were independent from each other, in a randomly determined sequence. Notably, unit activities right before the cue resided at an attractor state (Vyas et al., 2020). Using the library from Golub & Sussillo (2018), we confirmed that this constitutes a stable fixed point. For a given cue type, all trials followed the same trajectory (Fig. 2c; Fig. 9a, right in Appendix). Depending on whether reward was received, there were 6 trajectories to follow during the ITI period before receiving a new cue or unpredicted click (Fig. 9a, left in Appendix). On each trajectory, trials with different ITIs approached the fixed point such that the subsequent cue could trigger population activity to follow a fixed trajectory determined solely by that cue.

These observations also held for the multiple delays task (Fig. 3d, Fig. 9b in Appendix). Interestingly, for the multiple delays task, all seven trajectories resided on the same plane in the PCA space during the ITI period (Fig. 9c in Appendix).

### 3.3 RPEs AND VALUES IN THE OPERANT BANDIT TASK

When trained for the bandit task using average-reward RL, the model displayed adaptive choice behavior (Fig. 10 in Appendix) and reached an average trial-wise reward (0.61) well above chance level (0.5; the maximal level for an agent with complete knowledge of reward probabilities would be 0.67). The RPEs at the reward cue or omission (which occur at the "side-in" event) scaled inversely with recent reward rate (Fig. 4a, Fig. 11d in Appendix), resembling that observed for rats trained with the same task (Mohebi et al., 2019; 2024). Trial onset (light-on) also evoked (smaller) RPEs (Fig. 11a in Appendix); these scaled positively with reward rate, reflecting the greater expectation of upcoming reward when more recent trials had been rewarded.

### 3.4 POPULATION DYNAMICS IN THE OPERANT TASK

Unit activity right before side-in, displayed in the first two PCA axes, showed clustering according to both reward history (Fig. 4b, right) and left/right choices (Fig. 4c, right). This separation in state space was particularly obvious when considering different types of trial blocks. Unit activities at side-in for trials in three block types with high, medium, and low reward rates clustered into different regimes along the first PCA axis (PC1; Fig. 4b, left). In four other block types with clearly distinct left vs right reward probabilities (e.g., [0.9, 0.1], [0.1, 0.5]), the unit activities showed a clustering along the second PCA axis (PC2; Fig. 4c, left). This clustering was already apparent even before the trial start at light-on (Fig. 11b, c in Appendix). In this way, the network located its unit activity into corresponding dynamical regimes reflecting recent prior experience, and adaptively biased the choice to be made in the upcoming trial (Wang et al., 2018). This contrasted with the attractor dynamics observed for the conditioning tasks, where the trials were wholly independent from each other and maintaining a network state based on recent history would not be helpful.

## 4 DISCUSSION

In this work we compared RPEs from RNNs in simulations of behavioral tasks, with DA transients in the VS of rats performing those tasks. We conclude that training using average-reward RL can reproduce RPE-like features of VS DA across multiple tasks, and in some respects does so better than discounting with a very long time horizon ($\gamma$ very close to 1). This provides evidence that the prediction of future reward by brain circuits segregates away the component of value that is shared across states, as happens in average-reward RL (and also reward centering). Of course, the present work is only one, limited step towards understanding the brain processes of value estimation and the control of DA signals. DA release in VS is regulated by many factors including other neuromodulators, DA autoreceptors, and complex local circuits (Liu & Kaeser, 2019; Holly et al., 2024). Furthermore, striatal circuits also do not operate in isolation, but rather are components of broader networks involved in value estimation, including e.g., the frontal cortex (Wang et al., 2018) and other sub-brain structures like the amygdala (Averbeck & Costa, 2017).

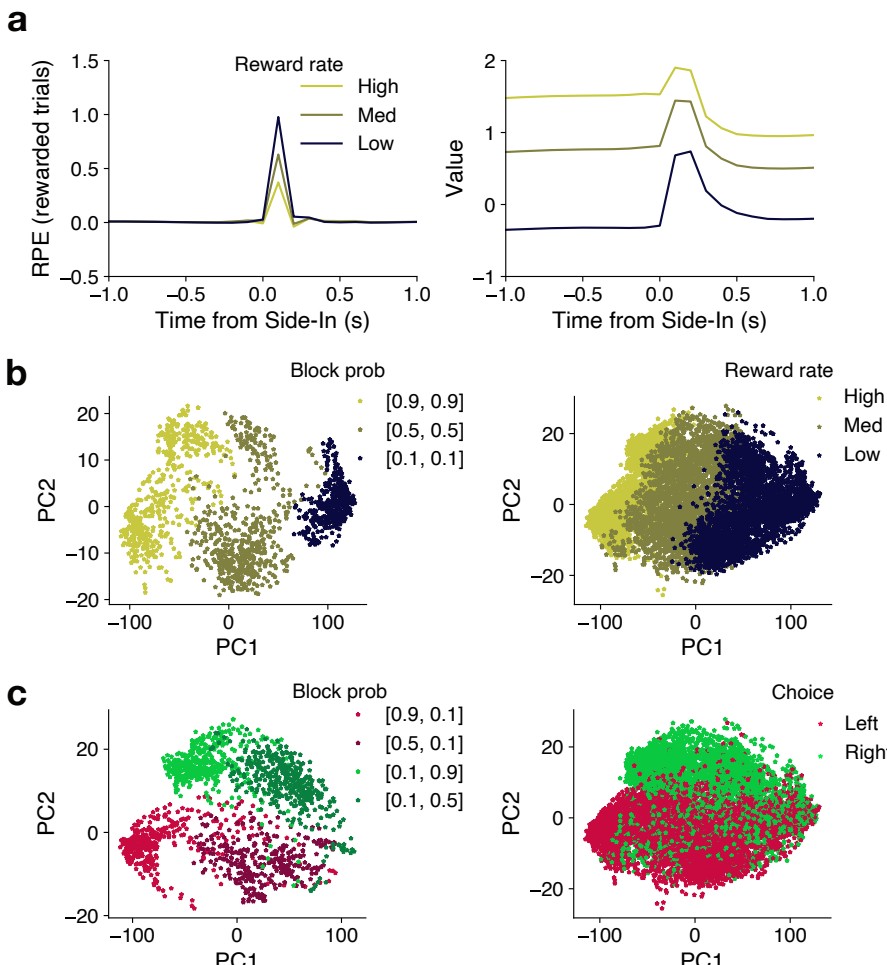

Figure 4: RPEs, values and population activity for the RNN model trained using average-reward RL in the bandit task. **a**. RPEs (left) and values (right) at the side-in event. **b**. Clustering of population activity at side-in reflecting reward history. Left, data from the last 20 trials in blocks with high ([0.9, 0.9]), medium ([0.5,0.5]) and low ([0.1, 0.1]) reward probabilities. Right, all trials with high, medium and low reward rates. **c**. Clustering of population activity at side-in reflecting choice bias. Left, data from the last 20 trials in each of the four block types in which one choice is better than the other, including left-better blocks ([0.9, 0.1], [0.5, 0.1]), and right-better blocks ([0.1,0.9], [0.1,0.5]). Right, all trials labeled with choice in the current trial. Data were from 200 blocks of evaluation.

DA has been implicated in both motivational and reinforcement processes (Dayan & Balleine, 2002; Berke, 2018). Tonic (slowly varying) DA has been argued to control motivation or vigor, and has been specifically suggested to encode a time-varying reward rate as part of optimizing time allocation (Niv et al., 2007). This continuously adjusted reward rate has been used in some implementations of average-reward RL (Daw & Touretzky, 2002). However, we used average-reward RL in a quite distinct way- for batch-level training of RNN models rather than an online, time-varying learning target or decision variable. We do make use of ongoing reward rate for analysis of RNN dynamics—finding that it is implicitly encoded in the RNN population state—but it is not a direct part of the RNN training process.

In the conditioning tasks both forms of RNN training (average-reward vs discounting with long-time-horizon; Fig. 7 in Appendix) resulted in a stable fixed point right before cue onset. This attractor may have been important for mimicking VS DA transients at cue onset—specifically the distinct and stereotyped responses to different cues, regardless of inter-trial-interval or prior trial history.

Attractors in the population dynamics of neuronal activities have been identified in various tasks, exhibiting not only point attractors but also more complex structures such as 1D and 2D attractors (Seung, 1996; Mante et al., 2013; Chaisangmongkon et al., 2017; Inagaki et al., 2019; Chaudhuri et al., 2019; Vyas et al., 2020; Finkelstein et al., 2021; Khona & Fiete, 2022; Langdon et al., 2023; Sorscher et al., 2023).Population dynamics and attractor structures have also been explored in recurrent networks trained on multiple tasks (Yang et al., 2019; Driscoll et al., 2022; Goudar et al., 2023; Turner & Barak, 2024). Another limitation of the present work is that we simulated each task using separate RNNs. It would be interesting to investigate how population dynamics track key decision variables when a single RNN is trained using average-reward RL to perform both conditioning and operant tasks.

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

# A  APPENDIX

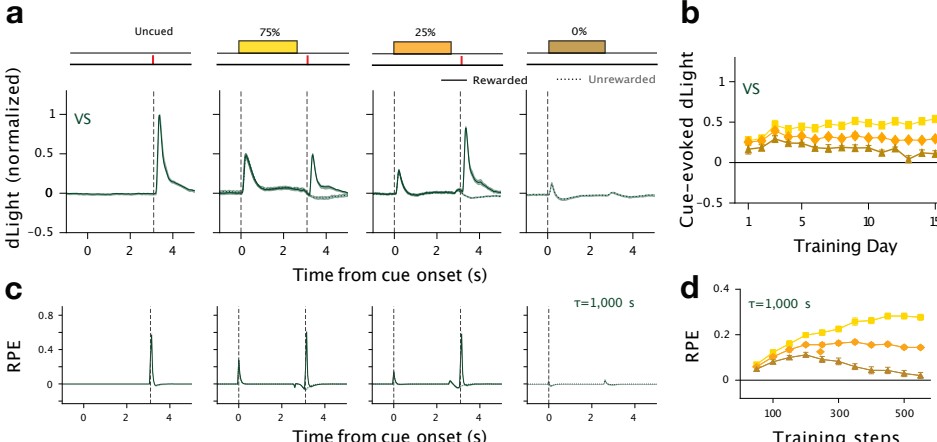

Figure 5: Comparison of experimental VS DA observations with results from an RNN using discounting with a long time horizon. Data adopted from Mohebi et al. (2024)). **a**. Top, cues and potential reward click times for different trial types. Bottom, DA transients in the VS, averaged over the last three days of training. **b**. Development of DA transients at cue onset over training. **c**. RPEs from the discount RL model for the VS with $\tau = 1000$ s at training step 500. Note that the discounting model RPE after the reward click is not smaller for the 75% cue (yellow) than for the 25% cue (orange), in contrast to the DA transients. **d**. Development of RPEs at cue onset over training.

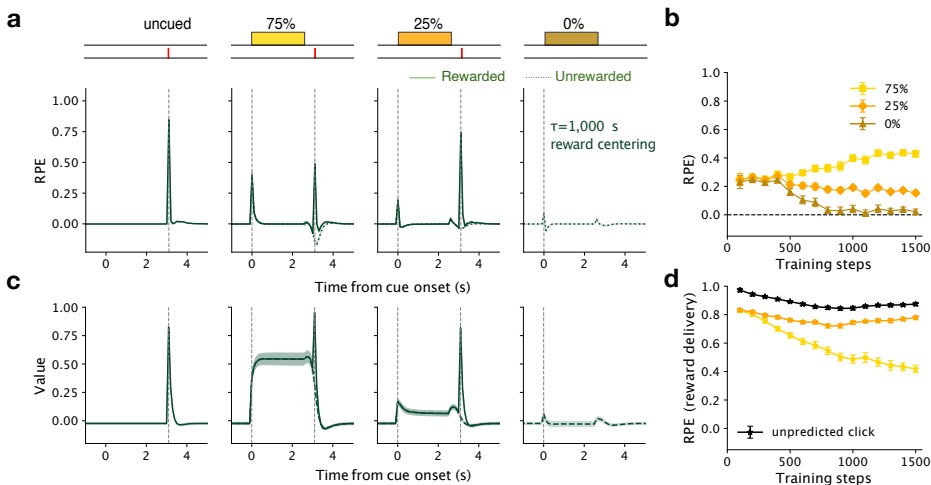

Figure 6: RPEs and values from an RNN trained with discount RL with a long time horizon utilizing reward centering. **a**. Top, cues and potential reward click times for different trial types. Bottom, RPEs for each trial type at training step 1000 from an RNN with $\tau = 1000$ s (corresponding to $\gamma = 0.9999$ with $dt = 0.1$ s) and reward centering. **b**. Development of RPEs at cue onset time for the three cues over training. **c**. Value functions for each trial type at training step 1000. **d**. Development of RPEs at reward click time in rewarded trials over training. Data were presented as mean $\pm$ s.e.m, averaged over 20 seeds with each run including 1000 trials.

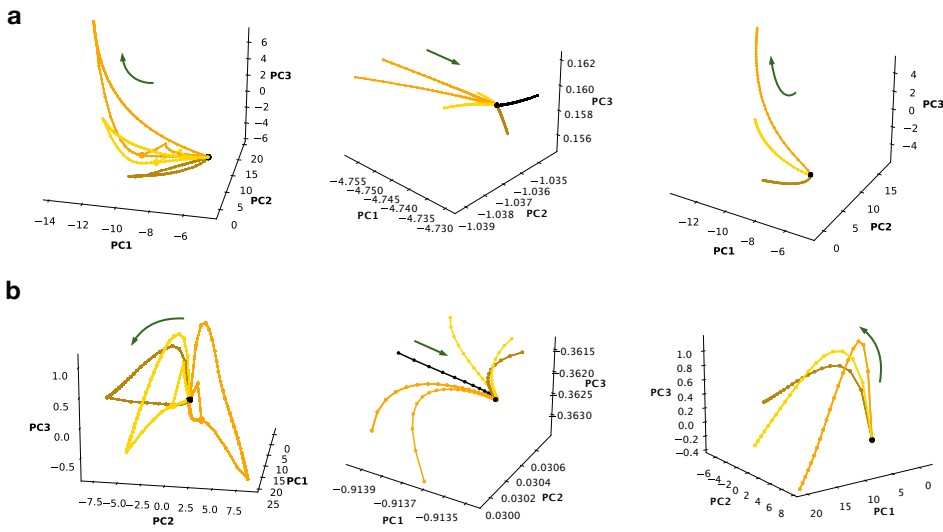

Figure 7: Population activity trajectories for units from RNNs with long time horizon. The time horizon used, $\tau = 1000$ s, corresponds to $\gamma = 0.9999$ with $dt = 0.1$ s. **a**. Trajectories of all trials from the probabilistic rewards task in PCA space for the whole trajectories (left), ITI periods, excluding the first 3 s (middle, 6 trajectories in total), and 2 s since cue onset (right) without reward centering. **b**. Same as **a** but with reward centering. The directions of arrows indicate time flow.

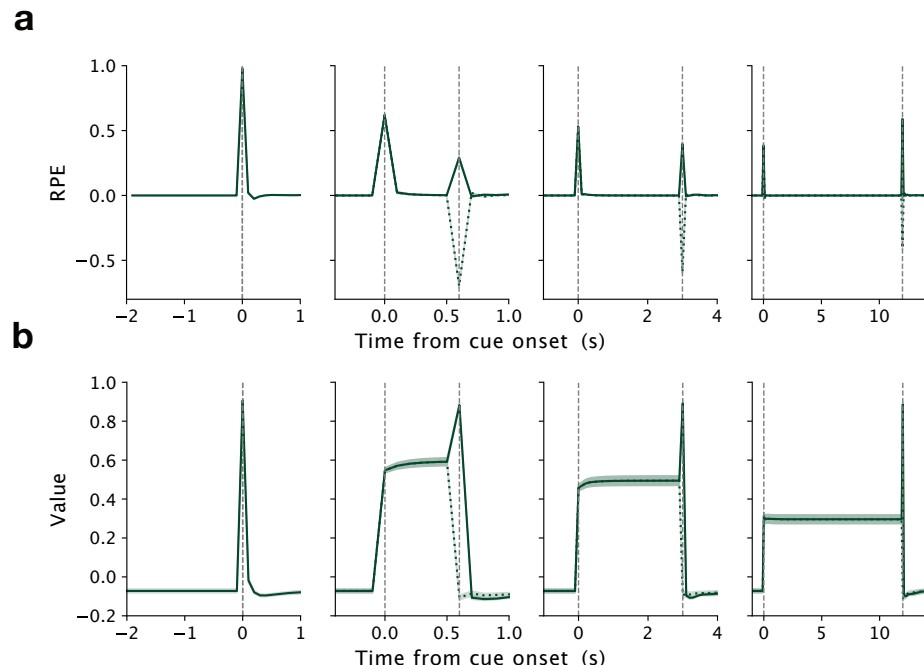

Figure 8: RPEs (**a**) and values (**b**) for the RNN model trained using average-reward RL in the multiple delays task. Data are presented as mean ± s.e.m from 10 seeds at training step 3000.

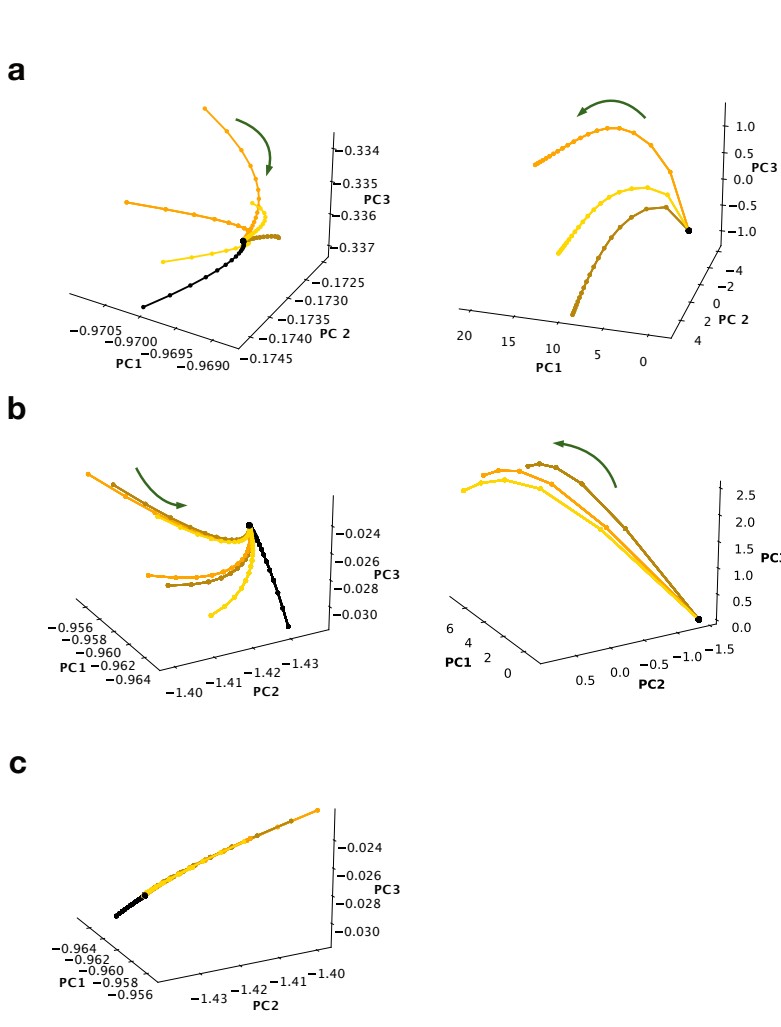

Figure 9: Population activity trajectories for units from RNNs trained using average-reward RL in the two conditioning tasks. **a**. Trajectories of all trials from the probabilistic rewards task in PCA space for ITI periods, excluding the first 3 s (left, 6 trajectories in total), and 2 s since cue onset (right). **b**. Same as **a** but for the multiple delays task. There are 7 trajectories in the left panel in total. **c**. The left panel of **b**, viewed from a different angle. The directions of arrows indicate time flow.

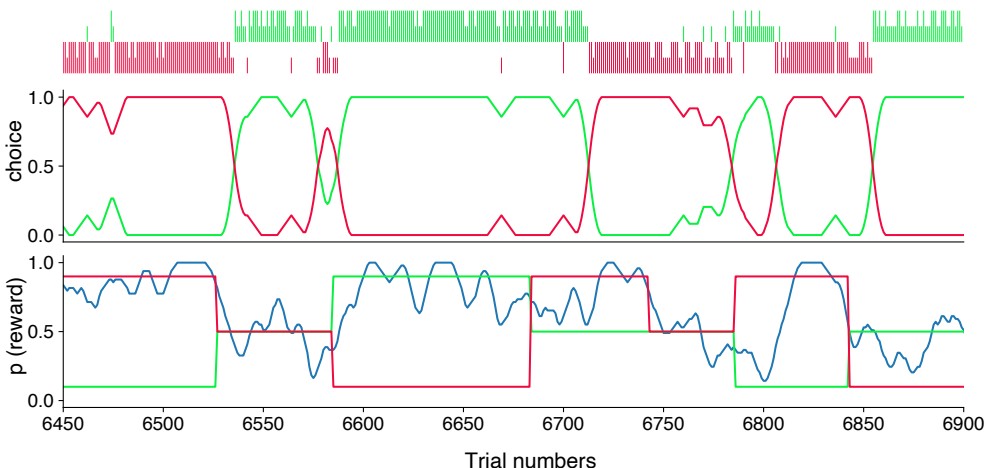

Figure 10: Example of behavioral choice adaptation from the RNN model trained with average-reward RL in the bandit task. Top, rewarded (long bars) and non-rewarded (short bars) trials for left (red) and right (green) choices. Middle and bottom panels show choosing probabilities and reward probabilities, respectively.

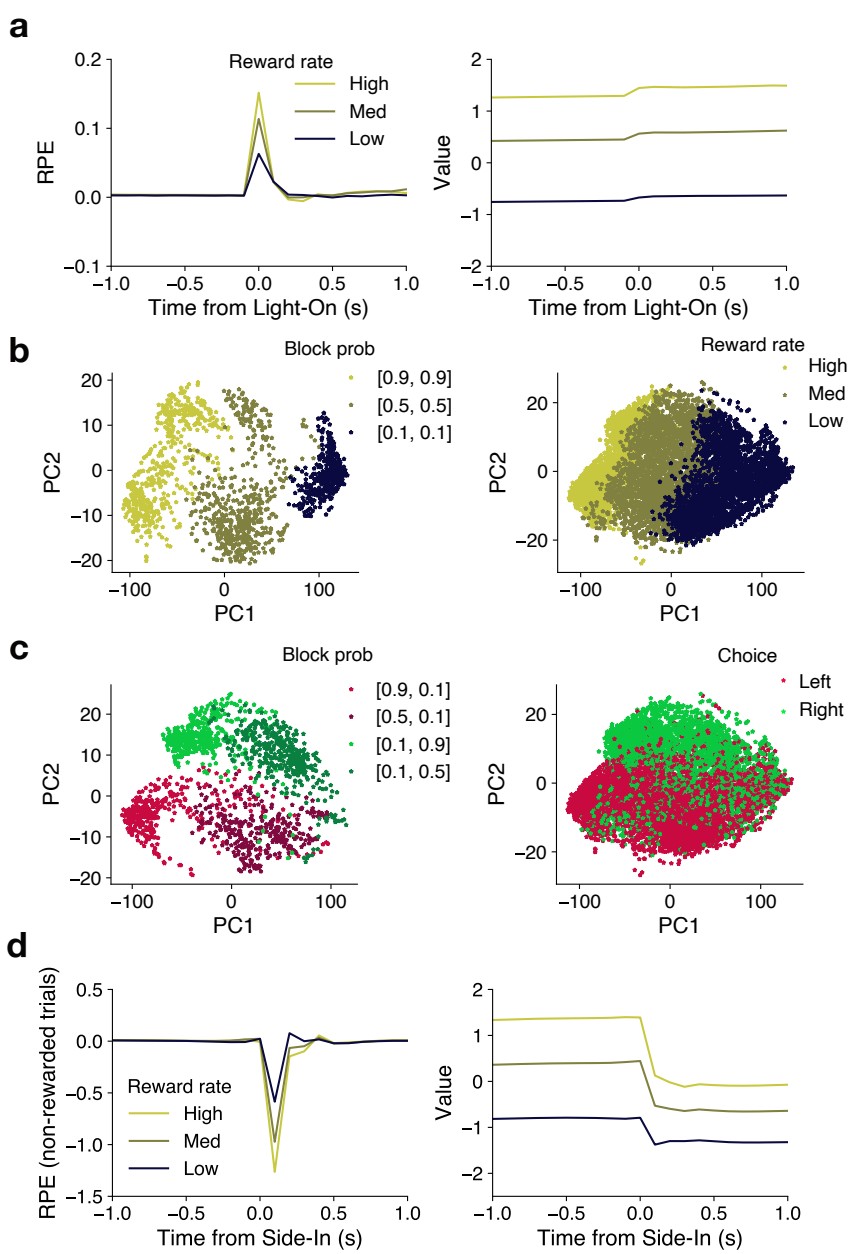

Figure 11: Additional results for RPEs, values and unit activities from the RNN model trained using average-reward RL in the bandit task. **a**. Dependence of RPEs (left) and values (right) on reward rate when aligned with the light-on event. **b**. Clustering of population activities at light-on reflecting reward history. Left, the last 20 trials in blocks with high ([0.9, 0.9]), medium ([0.5,0.5]) and low ([0.1, 0.1]) reward probabilities. Right, all trials with high, medium and low reward rates. **c**. Clustering of population activities at light-on reflecting choice preference. Left, the last 20 trials in four block types with clearly distinct reward probabilities, including left-better blocks ([0.9, 0.1], [0.5, 0.1]), and right-better block ([0.1,0.9], [0.1,0.5]). Right, population activity states for all trials, labeled by choice in the current trial. **d**. Dependence of RPEs (left) and values (right) on reward rate for non-rewarded trials when aligned with the side-in event. Data were from 200 blocks of evaluation.

