# OpenReview forum: "Dopamine transients in the ventral striatum provide evidence for average-reward reinforcement learning"
_ICLR.cc/2025/Conference — ICLR 2025 Conference Withdrawn Submission_

### Official Review · Reviewer_UQup · 2024-10-25

**Soundness:** 3
**Presentation:** 4
**Contribution:** 2
**Rating:** 5
**Confidence:** 4

**Summary:**

In this work, the Authors investigate an alternative explanation for dopamine signaling in the brain by comparing it with the predictions of R-learning. They show in a series of recordings from neuroscience experiments that R-learning can explain the dopamine activity in the brain.

**Strengths:**

Significance. The work takes upon an important topic that has gained a renewed interest in neuroscience. While the role of dopamine signaling has originally been attributed to value learning, other models and explanations for such signaling are being proposed in recent works. Current work extends this body of research by investigating the correspondence between dopamine signaling and R-learning.

Clarity. The text is remarkably well-written. I specifically appreciated the detailed explanation of the models. The vector figures are of high quality. The text is easy to read and thus accessible to the broad ICLR community.

Quality. The highlights of the quality aspect of this work are the multi-faceted investigation of the parallels between the teaching signal in R-learning and the dopamine signaling in the brain. This work establishes that, in the considered tasks, R-learning qualitatively explains the trends observed in the dopamine data.

Originality. While the parallels between the R-learning and the dopamine data have been drawn before, the original contribution of this work is in investigating this question using new tasks, new data, and RNNs.

**Weaknesses:**

Quality. While the similarity between the R-learning and the dopamine signaling has been established in this paper, the similarity between classical RL models and the dopamine signaling has not been ruled out. Indeed, such similarities have been claimed in prior works. It would be really interesting to see a similar analysis applied to classical RL paradigms, such as V-learning.

Originality. Along the same lines, the correspondence between the R-learning and the dopamine signaling has been shown in previous works, however, I am not aware of previous studies that managed to distinguish that from the classical RL paradigms. In that sense, the work in its current format does not add new arguments in favor of R-learning compared to V-learning.

Due to this weakness, I tend to provide a borderline negative recommendation for this work. I will revisit it if additional results are provided along these lines.

**Questions:**

See weaknesses

---

### Official Review · Reviewer_E8WZ · 2024-11-03

**Soundness:** 2
**Presentation:** 3
**Contribution:** 1
**Rating:** 3
**Confidence:** 3

**Summary:**

For this paper, the authors conducted two Pavlovian one operant conditioning tasks with mice and in simulation and compared the activations of simulated RNNs with readings of photometry sensors in the rat ventral striatum. While the dorsolateral and dorsomedial striatum's DA activations were succesfully modelled using short-term value learning of discounted rewards, the ventral striatum is believed to encode long-term averaged reward prediction errors (RPEs). The presented study finds similarities in the RPEs produced by the simulated networks and the rat brain recordings.

**Strengths:**

The papers premises are laid out concisely and the Related Work is addressed adequately: Relevant recent neuroscience studies as well as reinforcement learning literature such are forayed. The basic idea and research is interesting and the chosen route of investigation is original.

**Weaknesses:**

The presented study appears to lack novelty. No novel RL technique is introduced, but an existing algorithm is used to model very simple tasks and partially confirm findings from prior work. The paper further fails at communicating significance of the results and at placing them into the bigger picture of the current state of Reinforcement Learning Research.

**Questions:**

- How could the long-term RPE of the VS be combined with the discounted short-term RPEs of the other parts of the striatum to be used in an Actor-Critic style Reinforcement Learning algorithm?
- How does the library from Golub & Sussillo facilitate confirming a stable fixed point?
- The wording "with content" e.g. on Line 99 is unclear to me.
- You mention that you were using average-reward "for batch-level training of RNN models rather than an online, time-varying learning target or decision variable". Does this hamper the significance of the presented results? How would the results compare to running the simulation with a training batch-size of 1?

---

### Official Review · Reviewer_JQGU · 2024-11-04

**Soundness:** 3
**Presentation:** 3
**Contribution:** 2
**Rating:** 5
**Confidence:** 3

**Summary:**

This paper proposes the average-reward reinforcement learning as a model of dopamine release in the rat ventral striatum, as reported in Mohebi et al. (2019, 2024). The model reproduces the experimental data well, but the difference with the temporal discounting RL with a very large discount factor is minor, as expected.

**Strengths:**

This paper features experimental data from recent studies using fluorescent dopamine sensors and also models the tasks with up-to-date reinforcement learning algorithms.

**Weaknesses:**

The difference between the average reward and large discount factor models is not so clear. In the conclusion, it should be clearly stated that why discounting framework should be rejected, rather than vaguely writing "in some respects."

**Questions:**

In Figure 3, it was not clear to me why are the responses to the relayed reward cues decrease with the average reward RL. Maybe some more explanation is needed. And which parameter of the model specifies the reduction in the responses?
With a large discount factor, value functions can have high a baseline level, which makes value differences between actions hard to track. But in advantage functions can still show clear difference across actions, after subtraction of the state value function. How does the result with average reward compare with that of advantage learning?
How does the parameter \alpha for the update of average rewards affect the model performance?

---

### Note · Authors · 2024-11-22

**Comment:**

In light of further review and guidance from the senior author, we have decided to withdraw this manuscript from consideration in its current form. We appreciate the efforts of the reviewers and the conference organizers.

**Withdrawal Confirmation:**

I have read and agree with the venue's withdrawal policy on behalf of myself and my co-authors.